# Polygynous marriage union among Ghanaian Christian women: Socio-demographic predictors

Abdul Rauf Alhassan[1,2]*

1 Department of Surgery, Tamale Teaching Hospital, Tamale, Ghana, 2 Hasbi Research Consultancy, Tamale, Ghana

* Alhassana84@yahoo.com, abdul-rauf.alhassan@tth.gov.gh

## Abstract

### Introduction

Polygamy has declined in the last decade, but it is still prevalent in West African nations including Ghana even with the arrival of Christianity and colonists, which came to be recognized as a form of slavery that needed to be abolished.

### Aim

To analyze the determinants of polygyny among married Christian women in Ghana.

### Methods

Ghana Maternal Health Survey data was used for this study to do an analytic cross-section study. Data analysis was done using SPSS version 20. The association between dependent and independent variables was explored using chi-square and logistic regression. Statistical significance was set at p < 0.05.

### Results

The prevalence of Ghanaian Christian women's involvement in polygyny marriage union was 12.2%, the prevalence was higher (15.0%) among women of Anglican denomination, catholic denomination (13.9%), and the lowest (8.4%) prevalence recorded among those of Methodist denominations. The predictor factors identified include the age of the woman, history of education, type of place of residence, region, ethnicity, early sex initiation, and history of multiple unions.

### Conclusion

The prevalence of polygyny in this present study is high given the strict position the Christian religion has against polygyny. This study recommends that the pros and cons of polygyny are objectively looked at from a scientific point rather than a religious point of view.

**Data Availability Statement:** All dataset related to the findings of this study is available online at www.dhsprogram.com.

**Funding:** Funding for this study was completed by Ghana Organization for Maternal and Child Health

(GOMaCH). The funder had no role in study design, data collection and analysis, or preparation of the manuscript.

**Competing interests:** The authors have declared that no competing interests exist.

## Introduction

Polygamy is a type of marriage in which several spouses are involved. It can happen as polygyny (when a man has multiple wives at the same time). It can also be polyandry (when a woman has multiple husbands at the same time), or polygynandry (concurrent marriage of two or more wives to two or more husbands) [1]. The most common form of polygamy is polygyny and more than 80% of preindustrial societies had it [2]. Even though the global prevalence of polygyny is low, more than a third of the world's population lives in a community that allows it [3]. Polygyny has been practiced by various cultures throughout the world for many centuries. In most African countries, it has been an essential component of family law [2]. However, with the arrival of Christianity and colonists, it came to be recognized as a form of slavery that needed abolishment. As a result, its prevalence has been steadily decreasing for decades. Despite this, it is still more prevalent in Sub-Saharan Africa (SSA) than anywhere else [4]. The 'polygyny belt,' which stretches from Senegal in West Africa to Tanzania in East Africa, has the highest prevalence of polygyny in Africa [2]. According to another DHS report, polygyny accounts for 25% of all marriages in the Democratic Republic of the Congo (DRC), 47% in Sierra Leone, and 53% in The Gambia [5].

In a number of West and Central African nations, such as Burkina Faso (36%), Mali (34%), and Nigeria (28%), polygamy is prominent. Polygamy is permitted, at least in part, in these nations. Muslims in Africa are more likely than Christians to live in this kind of arrangement (25% vs. 3%), while in some nations the practice is also common among folk religion followers and nonreligious individuals. For instance, in Burkina Faso, polygamous homes are home to 24% of Christians, 40% of Muslims, and 45% of folk religion practitioners [2]. In Ghana, a study done using nationally representative data from the 2017/2018 Ghana Multiple Indicator Cluster Survey revealed the prevalence of polygyny to be 21.6% [6].

Demographic factors such as high infant and child mortality, high male mortality and out-migration, and potentially lethal male activities such as hunting and military combat contribute to an excess supply of women and a scarcity of men, which can promote polygyny [7]. Polygyny has become more common due to religion, particularly Mormonism and Islam. Age, place of residence, and household wealth all influence the prevalence of polygyny [8]. Other major factors influencing polygyny acceptance include culture and tradition. Polygyny is recommended as the solution to infertility and menopause in many African cultures. Polygyny increases the number of children available for domestic work, farming, and cattle herding in agricultural societies [4,9]. Polygyny is associated with an increased risk of infant and child mortality [10]. Children from polygynous families have poor health and die from malnutrition and HIV/AIDS [11]. Many studies have identified polygyny as one of the factors that influence early marriage, domestic violence, harmful traditional practices, and high fertility [12,13].

Polygamy has declined in the last decade, but it is still prevalent in West Africa. This ancient practice is encouraged by customary law and/or religious practices. Polygamy is also recognized and governed by civil law across most West African countries, allowing a man to marry up to four women in certain situations, including the financial capacity to support multiple wives and families [14]. In most cases, a polygamous union is limited to two women per couple. Six Countries in Africa (Benin, Cabo Verde, Côte d'Ivoire, Ghana, Guinea, and Nigeria) have civil codes that formally prohibit polygamy, but the laws are poorly applied. Other nations, such as Burkina Faso and Togo, recognize polygamous unions under contemporary civil law but only permit couples or men to be involved [14].

With the arrival of Christianity and colonists, it came to be recognized as a form of slavery that needed to be abolished [4]. Polygamy has been a source of consternation for Christian missionaries. In the past, Christian mission institutions in a broad sense have been unable to

reach an agreement on how to address this issue. In the highlands of New Guinea, for example, Roman Catholic and Lutheran missions condemn polygamy, refuse to baptize members of polygamous marriages, and demand that polygamous marriages be dissolved. They regard polygamy as a sin. Baptist and Methodist missions, but on the other hand, baptize those who entered a polygamous marriage before hearing the Gospel or, more particularly, before deciding to accept Christ. They do not regard polygamy to be a sin, but they believe it is not God's ideal. The view of all these missions is from the Bible [15]. This has motivated this current study to conduct a multilevel analysis of determinants of polygyny among married Christian women in Ghana.

## Materials and methods

Ghana is the study area. Ghana is a nation in West Africa that is bordered to the east by Togo, to the west by the Ivory Coast, to the south by the Gulf of Guinea and the Atlantic Ocean, and the north by Burkina Faso. The current population is 30.08 million [16]. Pentecostal/Charismatic Christians were found to make up the majority of Ghana's population and dwellings in the 2021 census, with a share of 31.6 percent. This resulted in a rise in population compared to the 2010 census year, to over 9.7 million people. Following closely behind was the Islamic world, with a national coverage of about 20%. Also, the percentage of people who did not practice any religion fell from 5.3 percent in the previous census year to just 1.1 percent [17].

This study is based on data from the 2017 Ghana Maternal Health Survey in an analytic cross-sectional survey design (GMHS). The Ghana Statistical Service (GSS) with technical help from the ICF's Demographic and Health Survey (DHS) program carried out the 2017 GMHS. The sampling frame was derived from Ghana's 2010 Population and Housing Census (PHC). Women between the ages of 15 and 49 years who were permanent residents of selected households or guests who stayed in selected households the night before the survey were eligible to participate. A multistage stratified cluster sampling technique was used to select study areas and households. The final report contains specifics about the survey procedures and questionnaires used [18].

The study included all survey's currently married Christian women participants (6393), and the study's primary dependent variable was polygyny marriage union. Demographic characteristics, premature sexual initiation, early marriage, and several union experiences were among the independent variables.

### Statistical analysis

SPSS Statistics for Windows, version 20.0, used for statistical analysis (IBM SPSS Statistics for Windows, Version 20.0. Armonk, NY: IBM Corp). Tables and figures are used to present the results of categorical variables using frequencies and percentages. The mean and standard deviation of continuous variables are used to represent the results. The chi-square test was used to determine the relationship between the dependent and independent variables. A binary logistics regression model was used to identify predictor variables of polygyny marriage unions for factors that showed a significant association at the bivariate level of analysis using chi-square. A p-value of less than 0.05 was used to define statistical significance.

### Ethical consideration

The ICF Institutional Review Board approved the protocol for the 2017 GMHS. Meanwhile, ethical approval was not required for this study because it involved a secondary analysis of a dataset without exposing the respondents' and their households' identities. Nonetheless,

permission to use the datasets in this study was obtained from ICF via the DHS program, and the data terms followed.

## Results

### Demographic characteristics of study participants

The majority (70.2%) of the study participants were 30 years and above, and about 53.7% of them were residing in rural areas. The Christian domination that dominated the (49.1%) study participants was a Pentecostal or charismatic group and the region with dominant (16.4%) representation was the Upper East region. Moreover, the ethnic group with the majority (40.0%) representation was Akan. About half (50.4%) had early sexual initiation, 29.7% were into child marriage, and 14.8% had had more than one union experience (Table 1).

### Prevalence of Christian women involved in a polygyny marriage

The prevalence of Ghanaian Christian women's involvement in polygyny marriage union was 12.2%, the prevalence was higher (15.0%) among women of Anglican denomination, catholic denomination (13.9%), and the lowest (8.4%) prevalence recorded among those of Methodist denomination (Fig 1). The majority (46.9%) of the women in the polygyny marriage union was first wives in terms of rank, and then 45.6% of them were second wives (Fig 2).

### Factors associated with Christian women involved in a polygyny marriage

Chi-square analysis revealed that all independent variables included in the study had a significant association with Christian women involved in polygyny marriage (Table 2). These variables were further modeled using a binary logistics regression model to identify predictor variables of Christian women involved in polygyny marriage.

Firstly, the age of the women predicted they're involved in polygyny marriage, those aged 30 years and above were 540% more likely to involve in a polygyny marriage union compared to those aged 15–19 years (AOR = 6.4, 95%, C.I = 2.7–14.9). Secondly, those without a history of education were 70% more likely to engage in polygyny marriage union compared to those with (AOR = 1.7, 95% CI = 1.4–2.0). In addition, rural residents were 120% more likely to engage in polygyny marriage compared to those urban residents (AOR = 2.2, 95% CI = 1.7–2.7). Northern regions were 70% more likely to engage in polygyny marriage unions compared to those of Western regions (AOR = 1.7, 95% CI = 1.1–2.6). Again, ethnicity also predicted women's involvement in polygyny marriage unions; those of Ewe tribes were 70% more likely to engage in polygyny marriage unions when compared to those of Akan tribes (AOR = 1.7, 95%, CI = 1.1–2.5). In addition, those of the Mole-Dagbani tribe were 200% more likely to engage in polygyny marriage unions when compared to those of the Akan tribes (AOR = 3.0, 95%, CI = 2.1–4.3). Again, those of the Gurma tribe were 270% more likely to engage in polygyny marriage unions when compared to those of the Akan tribe (AOR = 3.7, 95% CI = 2.5–5.5). Furthermore, the Christian denomination the women belong to predicted their involvement in the polygyny marriage union. Those of Methodist denominations were 90% more likely to engage in polygyny marriage union compared to those of Catholic denominations (AOR = 1.9, 95% CI = 1.2–3.0). In addition, those of Presbyterian denominations were 70% more likely to engage in polygyny marriage unions compared to those of Catholic denominations (AOR = 1.7, 95% CI = 1.1–2.6). Again, those of Pentecostal/charismatic denominations were 40% more likely to engage in polygyny marriage union compared to those of Catholic denominations (AOR = 1.4, 95% CI = 1.2–1.8). Nevertheless, women with a history of an earlier sexual debut were 50% more likely to engage in polygyny marriage union compared to

**Table 1. Demographic characteristics of study participant.**

| Frequency | | Percentage |
| --- | --- | --- |
| **Age of respondent** | | |
| 15–19 | 105 | 1.6% |
| 20–24 | 531 | 8.3% |
| 25–29 | 1134 | 17.7% |
| 30 and above | 4623 | 72.3% |
| **Ever attended school** | | |
| Yes | 4486 | 70.2% |
| No | 1907 | 29.8% |
| **Type of place of residence** | | |
| Urban | 2962 | 46.3% |
| Rural | 3431 | 53.7% |
| **Christian denomination** | | |
| Catholic | 1419 | 22.2% |
| Anglican | 61 | 1.0% |
| Methodist | 336 | 5.3% |
| Presbyterian | 367 | 5.7% |
| Pentecostal/charismatic | 3139 | 49.1% |
| Other Christian | 1071 | 16.8% |
| **Region** | | |
| Western | 697 | 10.9% |
| Central | 414 | 6.5% |
| Greater Accra Accra | 667 | 10.4% |
| Volta | 282 | 4.4% |
| Eastern | 667 | 10.4% |
| Ashanti | 770 | 12.0% |
| Brong ahafo | 544 | 8.5% |
| Northern | 689 | 10.8% |
| Upper east | 1046 | 16.4% |
| Upper west | 617 | 9.7% |
| **Ethnicity** | | |
| Akan | 2557 | 40.0% |
| Ga/Dangme | 303 | 4.7% |
| Ewe | 656 | 10.3% |
| Guan | 157 | 2.5% |
| Mole-dagbani | 1856 | 29.0% |
| Grusi | 267 | 4.2% |
| Gurma | 532 | 8.3% |
| Mande | 27 | 0.4% |
| Other | 38 | 0.6% |
| **Child marriage** | | |
| No | 4496 | 70.3% |
| Yes | 1897 | 29.7% |
| **Early sex initiation** | | |
| No | 3169 | 49.6% |
| Yes | 3220 | 50.4% |
| **In union more than once** | | |
| Only once | 5445 | 85.2% |
| More than once | 948 | 14.8% |

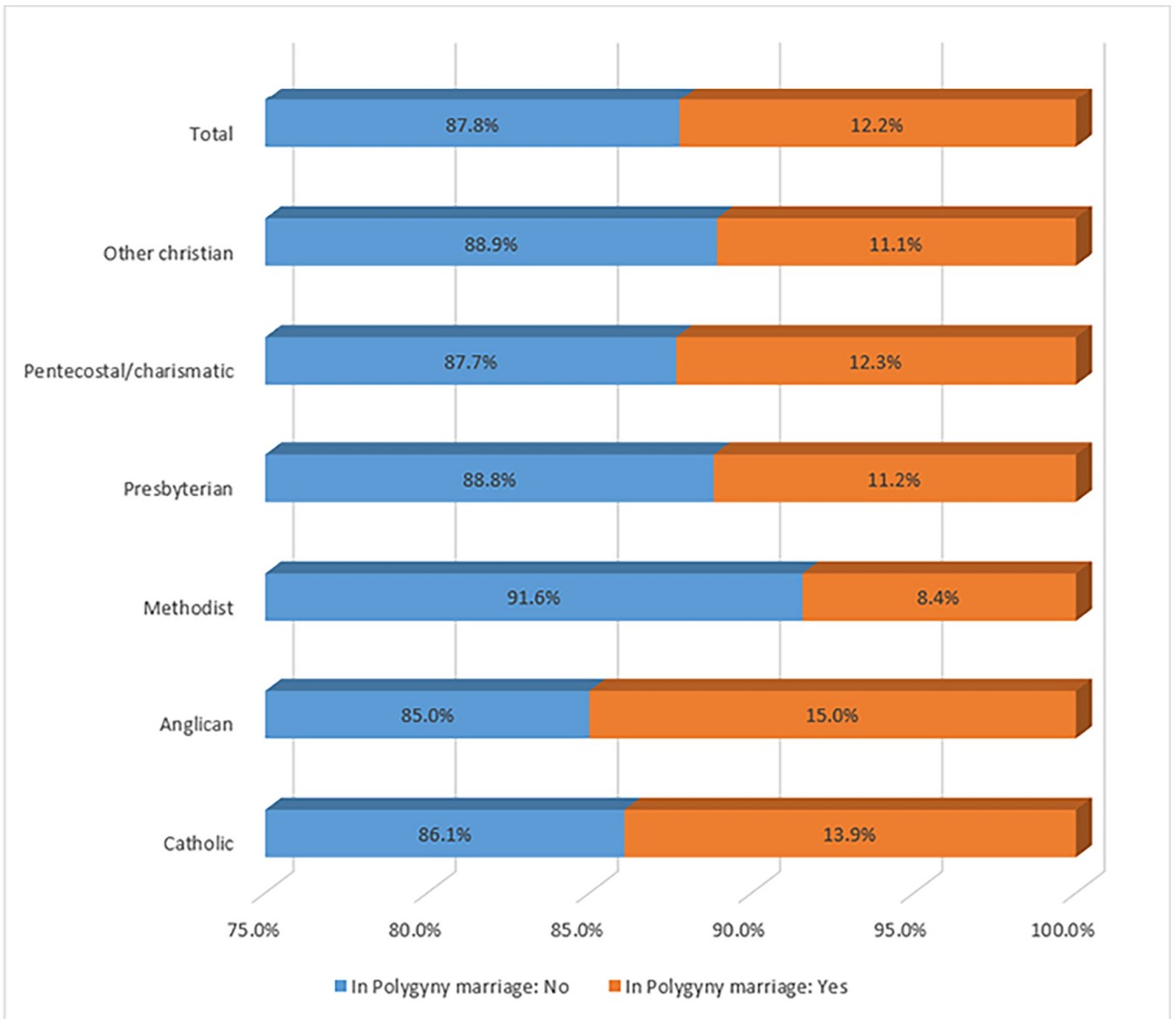

$X^2=10.412, P=0.064$

**Fig 1. Prevalence of Christian women involved in a polygyny marriage.**

those without (AOR = 1.5, 95% CI = 1.2–1.8). Lastly, women with a history of in-union more than once were 80% more likely to engage in polygyny marriage union compared to those without (AOR = 1.8, 95% CI = 1.5–2.2) (Table 3).

## Discussion

According to DHS report, polygyny accounts for 25% of all marriages in the Democratic Republic of the Congo (DRC), 47% in Sierra Leone, and 53% in Gambia [5]. In Ghana, the prevalence is 21.6% [6]. It became known as a sort of slavery that required abolition with the coming of Christianity and colonists. Its prevalence has been steadily declining as a result for

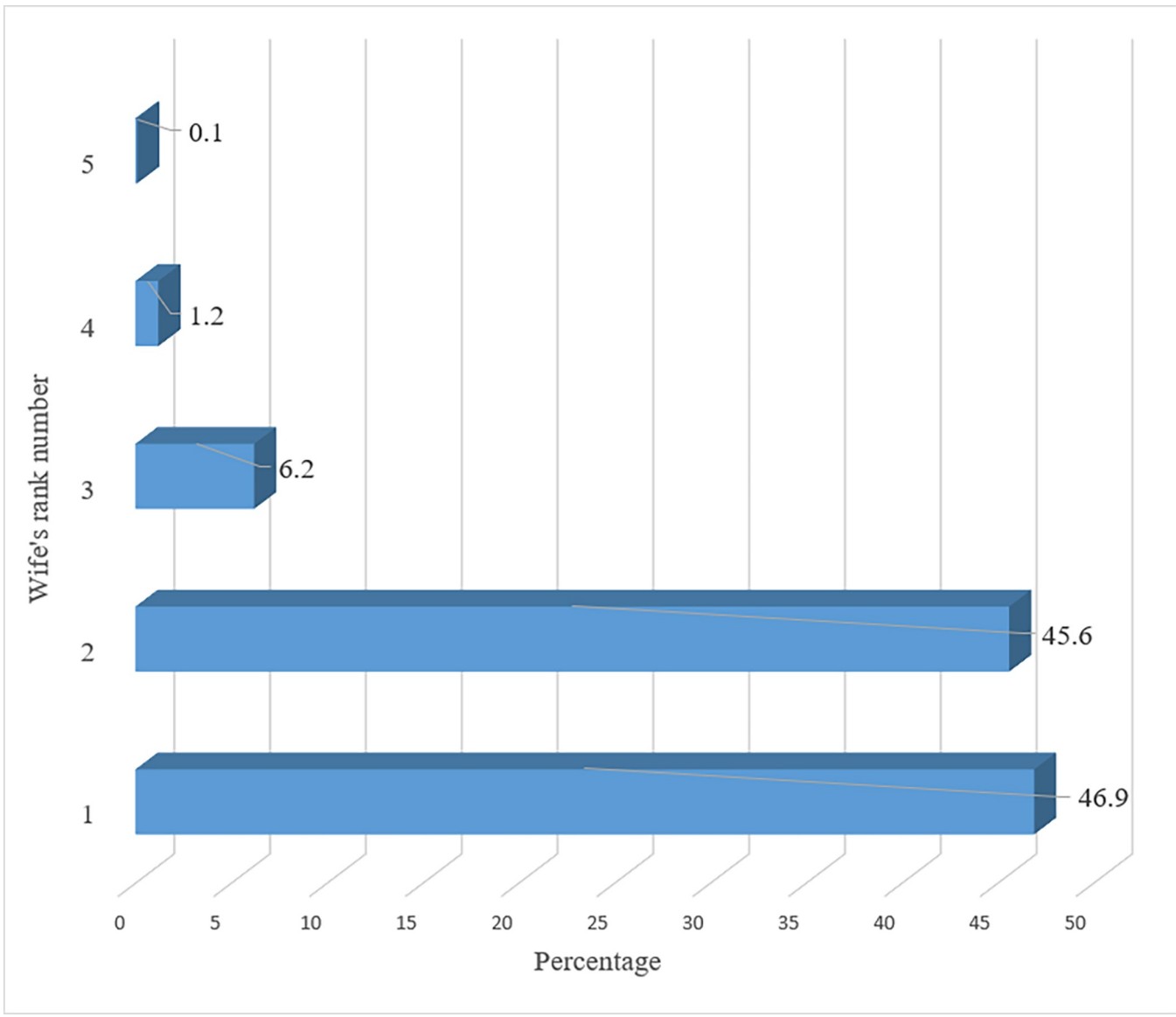

**Fig 2. The rank of Christian women involved in a polygyny marriage.**

decades. The region of Sub-Saharan Africa (SSA) continues to have the highest prevalence despite this [4]. This is evidenced in this present study, as polygyny was recorded to be 12.2% among Christians compared to 21.6% for the whole Ghana population, which included Muslims and traditional believers who accept the practice [6]. In proportion this is similar to the situation in Burkina Faso (40.0% for Muslims and 24.0% for Christians), but almost reverse with the case of Chad, where Christians (21.0%) were more likely than Muslims (10.0%) to be in polygamous household [2]. In most cases, a polygamous union is limited to two women per couple [14]. This explains the findings of this current study in which the majority were first wives and second wives in terms of wife's rank.

Roman Catholic and Lutheran missions have in the past denounced polygamy, refused to baptize individuals in polygamous unions, and requested that polygamous unions be dissolved. They consider polygamy to be sinful. Nonetheless, those who entered a polygamous

**Table 2. Chi-square analysis of factors associated with polygyny marriage relations among Christian women of Ghana.**

| | In Polygyny marriage | | Test statistics | |
| --- | --- | --- | --- | --- |
| | No | Yes | | |
| **Age of respondent** | | | | |
| 15–19 | 97 | 6 | Chi-square | 67.881 |
| 20–24 | 490 | 39 | Sig. | .000* |
| 25–29 | 1053 | 74 | | |
| 30 and above | 3945 | 659 | | |
| **Ever attended school** | | | | |
| Yes | 4161 | 299 | Chi-square | 423.851 |
| No | 1424 | 479 | Sig. | .000* |
| **Type of place of residence** | | | | |
| Urban | 2799 | 144 | Chi-square | 274.426 |
| Rural | 2786 | 634 | Sig. | .000* |
| **Region** | | | | |
| Western | 638 | 54 | Chi-square | 340.409 |
| Central | 377 | 34 | Sig. | .000* |
| Greater Accra | 641 | 23 | | |
| Volta | 239 | 42 | | |
| Eastern | 632 | 33 | | |
| Ashanti | 726 | 38 | | |
| Brong Ahafo | 482 | 58 | | |
| Northern | 505 | 183 | | |
| Upper east | 849 | 193 | | |
| Upper west | 496 | 120 | | |
| **Ethnicity** | | | | |
| Akan | 2417 | 126 | Chi-square | 410.767 |
| Ga/Dangme | 286 | 15 | Sig. | .000* |
| Ewe | 587 | 64 | | |
| Guan | 144 | 12 | | |
| Mole-Dagbani | 1481 | 369 | | |
| Grusi | 241 | 26 | | |
| Gurma | 370 | 160 | | |
| Mande | 25 | 2 | | |
| Other | 34 | 4 | | |
| **Child marriage** | | | | |
| No | 4022 | 449 | Chi-square | 66.857 |
| Yes | 1563 | 329 | Sig. | .000* |
| **Early sex initiation** | | | | |
| No | 2889 | 265 | Chi-square | 85.604 |
| Yes | 2692 | 513 | Sig. | .000* |
| **In union more than once** | | | | |
| Only once | 4805 | 613 | Chi-square | 28.323 |
| More than once | 780 | 165 | Sig. | .000* |

marriage before hearing the Gospel or, more specifically, before genuinely deciding to accept Christ, are baptized by Baptist and Methodist missionaries. Although they do not see polygamy as a sin, they do think it is contrary to God's will. All of these missions are viewed from a biblical perspective [15]. However, this present study revealed a higher prevalence among women

**Table 3. Binary logistics analysis of predictor factors of polygyny marriage relations among Christian women of Ghana.**

| | B | Sig. | AOR | 95% C.I. for AOR | |
|---|---|---|---|---|---|
| | | | | Lower | Upper |
| **Age of respondent** | | | | | |
| 15–19 | | | Ref | | |
| 20–24 | .597 | .195 | 1.816 | .737 | 4.474 |
| 25–29 | .763 | .088 | 2.145 | .894 | 5.148 |
| 30 and above | 1.849 | ≤ 0.001 | 6.354 | 2.713 | 14.883 |
| **Ever attended school** | | | | | |
| Yes | | | Ref | | |
| No | .511 | ≤ 0.001 | 1.666 | 1.365 | 2.033 |
| **Type of place of residence** | | | | | |
| Urban | | | Ref | | |
| Rural | .769 | ≤ 0.001 | 2.157 | 1.728 | 2.692 |
| **Christian denomination** | | | | | |
| Catholic | | | Ref | | |
| Anglican | .349 | .397 | 1.418 | .632 | 3.182 |
| Methodist | .616 | .011 | 1.851 | 1.152 | 2.972 |
| Presbyterian | .536 | .010 | 1.709 | 1.135 | 2.574 |
| Pentecostal/charismatic | .367 | ≤ 0.001 | 1.444 | 1.159 | 1.800 |
| Other Christian | .180 | .209 | 1.197 | .904 | 1.584 |
| **Region** | | | | | |
| Western | | | Ref | | |
| Central | .303 | .204 | 1.354 | .849 | 2.161 |
| Greater Accra | -.243 | .387 | .784 | .452 | 1.361 |
| Volta | .426 | .115 | 1.531 | .901 | 2.600 |
| Eastern | -.435 | .073 | .647 | .402 | 1.041 |
| Ashanti | -.310 | .173 | .733 | .469 | 1.146 |
| Brong Ahafo | -.076 | .728 | .927 | .605 | 1.420 |
| Northern | .538 | .012 | 1.712 | 1.128 | 2.598 |
| Upper east | .358 | .092 | 1.430 | .943 | 2.169 |
| Upper west | .232 | .307 | 1.261 | .808 | 1.967 |
| **Ethnicity** | | | | | |
| Akan | | | Ref | | |
| Ga/Dangme | .229 | .451 | 1.258 | .693 | 2.281 |
| Ewe | .527 | .010 | 1.693 | 1.134 | 2.530 |
| Guan | .072 | .830 | 1.075 | .557 | 2.075 |
| Mole-Dagbani | 1.089 | ≤ 0.001 | 2.971 | 2.072 | 4.260 |
| Grusi | .434 | .109 | 1.544 | .907 | 2.626 |
| Gurma | 1.313 | ≤ 0.001 | 3.719 | 2.495 | 5.542 |
| Mande | .008 | .991 | 1.008 | .225 | 4.520 |
| Other | .809 | .156 | 2.245 | .734 | 6.865 |
| **Child marriage** | | | | | |
| No | | | Ref | | |
| Yes | .098 | .317 | 1.103 | .911 | 1.335 |
| **Early sex initiation** | | | | | |
| No | | | Ref | | |
| Yes | .389 | ≤ 0.001 | 1.476 | 1.217 | 1.789 |
| **In union more than once** | | | | | |

*(Continued)*

**Table 3.** (Continued)

|  | B | Sig. | AOR | 95% C.I. for AOR | |
| --- | --- | --- | --- | --- | --- |
|  |  |  |  | Lower | Upper |
| **Age of respondent** |  |  |  |  |  |
| Only once |  |  | Ref | | |
| More than once | .583 | ≤ 0.001 | 1.792 | 1.451 | 2.213 |

of Anglican and catholic denominations, and the lowest prevalence recorded among those of Methodist denominations. Meanwhile, further multiple variable analyses revealed that those of Methodist denominations were more likely to engage in polygyny marriage unions compared to those of Catholic denominations. In addition, those of Presbyterian denominations were more likely to engage in polygyny marriage unions compared to those of Catholic denominations. Again, those of Pentecostal/charismatic denominations were more likely to engage in polygyny marriage unions compared to those of Catholic denominations.

Age may influence the prevalence of polygyny [8]. Many studies have identified polygyny as one of the factors that influence early marriage [12,13]. However, in this present study, child marriage did not predict polygyny among women, this further explained with those aged 30 years, and above are more likely, to involve in a polygyny marriage union compared to those aged 15–19 years. This study's finding is not different from the results of all demographic and Health Surveys of sub-Saharan African countries conducted since 2000 for 22 countries in which older age was positively associated with polygyny in each country result [19].

Place of residence, level of education, and household wealth all influence the prevalence of polygyny [6]. Poverty and education have a symbiotic relationship. This is because education provides knowledge and skills that lead to higher wages [20]. Due to the increasing incidence of poverty in rural Sub-Saharan Africa, girls are vulnerable to very few alternative income opportunities other than inside the bounds of marriage, which can result in very rapid marriage transactions in both families [21]. These relationships were repeated in this present study; those rural residents were more likely to engage in polygyny marriage compared to those urban residents. In addition, those without a history of education were more likely to engage in polygyny marriage unions compared to those with.

More so, the southern regions of Ghana are more urbanized than the northern regions of Ghana [22]. This explains why those of the northern region were more likely to engage in polygyny marriage unions compared to those of the Western region (southern region). This is a further clarification of why women of ethnic groups in northern regions were more likely to engage in polygyny marriage unions when compared to those of the Akan tribe of southern Ghana. Those of the Mole-Dagbani tribe were twice more likely to engage in polygyny marriage unions when compared to those of the Akan tribe. Again, those of the Gurma tribe were almost three times more likely to engage in polygyny marriage unions when compared to those of Akan.

Nevertheless, women with a history of an earlier sexual debut were more likely to engage in polygyny marriage unions compared to those without. In a recent study, the marriage status of women was associated with a history of earlier sexual debut [23]. Maybe women with early exposure to sex will prefer polygynous marriage instead of sticky to promiscuous sex. Lastly, women with a history of being in a union more than once were more likely to engage in polygyny marriage unions compared to those without. This means that once ever married, women prefer polygynous marriage to be single and promiscuous sex.

This study is not without limitations as not all factors including economic factors were explored in this study. However, the strength of this study is that the dataset employed is the national presentative survey by demographic and health survey.

## Conclusion

The prevalence of polygyny in this present study is high given the strict position the Christian religion has against polygyny. The predictor factors identified include the age of the woman, history of education, type of place of residence, region, ethnicity, early sex initiation, and history of multiple unions.

## Author Contributions

**Conceptualization:** Abdul Rauf Alhassan.

**Data curation:** Abdul Rauf Alhassan.

**Formal analysis:** Abdul Rauf Alhassan.

**Funding acquisition:** Abdul Rauf Alhassan.

**Methodology:** Abdul Rauf Alhassan.

**Resources:** Abdul Rauf Alhassan.

**Software:** Abdul Rauf Alhassan.

**Supervision:** Abdul Rauf Alhassan.

**Validation:** Abdul Rauf Alhassan.

**Visualization:** Abdul Rauf Alhassan.

**Writing – original draft:** Abdul Rauf Alhassan.

**Writing – review & editing:** Abdul Rauf Alhassan.

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
