## [Decision Letter · Decision Letter 0]

6 Nov 2022

PONE-D-22-26419Polygynous Marriage Union among Ghanaian Christian Women: Socio-demographic PredictorsPLOS ONE

Dear Dr. Alhassan,

Thank you for submitting your manuscript to PLOS ONE. After careful consideration, we feel that it has merit but does not fully meet PLOS ONE’s publication criteria as it currently stands. Therefore, we invite you to submit a revised version of the manuscript that addresses the points raised during the review process

The references should be reviewed. I believe that at least 70 percent of the references should not be older than five years. There are very old references in the article.

There are many grammatical errors in the article, and there is a need to involve someone proficient in the English language to review the paper before sending the corrected version back to Plos One Journal.

We look forward to receiving your revised manuscript.

Kind regards,

Innocent Ijezie Chukwuonye, MBBS, FMCP (Internal Medicine)

Academic Editor

PLOS ONE

2. One of the noted authors is a group or consortium Hasbi Research Consultancy (HasbiRC). In addition to naming the author group, please list the individual authors and affiliations within this group in the acknowledgments section of your manuscript. Please also indicate clearly a lead author for this group along with a contact email address.

4. Please ensure that you refer to Figure 2 in your text as, if accepted, production will need this reference to link the reader to the figure.

5. We note you have included a table to which you do not refer in the text of your manuscript. Please ensure that you refer to Table 1 in your text; if accepted, production will need this reference to link the reader to the Table.

Reviewers' comments:

Reviewer's Responses to Questions

**Comments to the Author**

1. Is the manuscript technically sound, and do the data support the conclusions?

Reviewer #1: Yes

Reviewer #2: Yes

2. Has the statistical analysis been performed appropriately and rigorously? 

Reviewer #1: I Don't Know

Reviewer #2: Yes

3. Have the authors made all data underlying the findings in their manuscript fully available?

Reviewer #1: Yes

Reviewer #2: Yes

4. Is the manuscript presented in an intelligible fashion and written in standard English?

Reviewer #1: No

Reviewer #2: Yes

5. Review Comments to the Author

Reviewer #1: Polygynous Marriage Union among Ghanaian Christian Women: Socio-demographic Predictors

Background

• I would have liked to see more on implications of polygyny socially, demographically, and with respect to sexual and reproductive health.

• The author should improve upon the language throughout the document.

• If feasible more recent literature should be used [examples of old literature: Holst, 1967; Okorje, 1994; Strassmann, 1997].

Methods

Were demographic and health survey data for Ghana used for analysis? If so, were weights applied during data analysis?

Results

The “majority” apply to over 50%. Please see for instance the section on “Prevalence of Christian women involved in a polygyny marriage”.

Discussion

Discuss the prevalence of polygyny and compare it with Islam and Christians countries in the region.

This section requires significant improvement. Findings and literature are presented again without modification or appropriate discussion. For instance see the first and second paragraphs of the discussion.

The explanation of the association between age and polygyny is not clear. “Age may influence the prevalence of polygyny (Okorje, 1994; Strassmann, 1997). Many studies have identified polygyny as one of the factors that influence early marriage (Ahinkorah, 2021; Gaffney-Rhys, 2012).” What were the author’s assumptions?

In this case polygyny appears to be associated with low socio-economic status (rural, low levels of education .... ) the results should be discussed further rather than just presenting the literature and findings.

Ethnicity is closely associated with culture. What are some of the specific cultural issues among the ethnic groups that could be linked to polygyny?

What are the implications of this: “Ghana's male-to-female ratio was 102.79 males per 100 females in 2020, up from 102.62 males per 100 females in 2015, representing a 0.16 percent increase (Index mundi, 2021).” How does this relate to the study? Important but please provide an explanation.

“Lastly, women with a history of being in union more than once were more likely to engage in polygyny marriage unions compared to those without. This means that once married, women prefer polygynous marriage to being single and promiscuous sex.” Do the women “prefer” or “resort” to polygynous marriages?

Limitations

If this is demographic survey data, the author should consider the variables: wealth quintile and occupation. The author should explore the dataset and include the variable in the analysis.

Reviewer #2: Abstract

The abstract is a true reflection of the overall study as presented.

Introduction

Literature review is satisfactory and is focused and aligned with the topic and objective.

The justification for the study was satisfactory.

Methodology

The authors should have given a brief socio-demographic description of Ghana, since it is national study, highlighting religious background of the country.

Ethical considerations for the study was adequate.

Results

The analysis and findings are in tandem with objective of the study.

The presentation of the results is not satisfactory; the percentages should be included in table 2.

However, the variables in the tables should be placed directly above the subcategories, to enhance readability. For instance.

Factors associated with polygyny

Age of respondent

15-19

20-24

25-29

Discussions/Conclusions

Page 11 paragrah 2, was repeated verbatim in page 15 paragraph 2.

Since the study population were Christians, the authors should have included comparison with findings among Christians elsewhere or other religious groups in addition to Ghana’s general population.

Otherwise, the discussion was satisfactory and was based on the findings from the analysis.

References

Most of the references are relevant and up to date.

6. PLOS authors have the option to publish the peer review history of their article (what does this mean?). If published, this will include your full peer review and any attached files.

Reviewer #1: No

Reviewer #2: **Yes: **Ugochukwu Onyeonoro

---

## [Author Response · Author response to Decision Letter 0]

3 Apr 2023

Academic editor comments

Comment: 

The references should be reviewed to at least 70% of references not older five years.

 Action:

The references were reviewed to at least 70% of references not older five years.

Comment:

There many grammatical errors in the article.

Actions:

Grammatical errors in the article attended to.

Reviewer’s comments

Comment (Methodology): 

Highlight religious background of Ghana

Actions:

Religious background of Ghana highlighted under Materials and Methods.

Comment (Results): 

Reformat tables 

Actions:

Comment (Discussion): 

1. Sentence copied verbatim from on paragraph to the other.

2. Include comparison with Christian study somewhere.

Actions:

1. Verbatim copied sentences edited

2. Christian studies somewhere compared in the discussion.

---

## [Editor Report · Decision Letter 1]

5 Apr 2023

PONE-D-22-26419R1Polygynous Marriage Union among Ghanaian Christian Women: Socio-demographic PredictorsPLOS ONE

Dear Dr. Alhassan,

Thank you for submitting your manuscript to PLOS ONE. After careful consideration, we feel that it has merit but does not fully meet PLOS ONE’s publication criteria as it currently stands. Therefore, we invite you to submit a revised version of the manuscript that addresses the points raised during the review process.

Your method of referencing did not adhere to that recommended by PLOS journals, and in addition, they are not numbered. Please read the method of referencing recommended by the PLoS One journal on the relevant page on our website, and make the appropriate corrections.

We look forward to receiving your revised manuscript.

Kind regards,

Innocent Ijezie Chukwuonye, MBBS, FMCP(Internal Medicine)

Academic Editor

PLOS ONE
---

## [Author Response · Author response to Decision Letter 1]

7 Apr 2023

Editor’s comments

Comments: Your method of referencing did not adhere to that recommended by PLOS journals, and in addition, they are not numbered. Please read the method of referencing recommended by the PLoS One journal on the relevant page on our website, and make the appropriate corrections.

Author’s Response

Responses: The referencing styles updated to Vancouver using Microsoft office referencing.

---

## [Editor Report · Decision Letter 2]

17 Apr 2023

Polygynous Marriage Union among Ghanaian Christian Women: Socio-demographic Predictors

PONE-D-22-26419R2

Dear Dr. Alhassan,

We’re pleased to inform you that your manuscript has been judged scientifically suitable for publication and will be formally accepted for publication once it meets all outstanding technical requirements.

Kind regards,

Innocent Ijezie Chukwuonye, MBBS, FMCP(Internal Medicine)

Academic Editor

PLOS ONE
---

## [Editor Report · Acceptance letter]

19 Apr 2023

PONE-D-22-26419R2 

Polygynous Marriage Union among Ghanaian Christian Women: Socio-demographic Predictors 

Dear Dr. Alhassan:

I'm pleased to inform you that your manuscript has been deemed suitable for publication in PLOS ONE. Congratulations! Your manuscript is now with our production department. 

Kind regards, 

on behalf of

Dr. Innocent Ijezie Chukwuonye 

Academic Editor

PLOS ONE